# Rubber Tree Crown Segmentation and Property Retrieval Using Ground-Based Mobile LiDAR after Natural Disturbances

**Ting Yun [1,2,†], Kang Jiang [1,2,†], Hu Hou [1,2,†], Feng An [3], Bangqian Chen [3], Anna Jiang [1], Weizheng Li [4] and Lianfeng Xue [1,2,*]**

[1] School of Information Science and Technology, Nanjing Forestry University, Nanjing 210037, China; yunting@njfu.edu.cn (T.Y.); jiangkang@njfu.edu.cn (K.J.); houhu@njfu.edu.cn (H.H.); annajiang@njfu.edu.cn (A.J.)

[2] Co-Innovation Centre for Sustainable Forestry in Southern China, Nanjing Forestry University, Nanjing 210037, China

[3] Danzhou Investigation and Experiment Station of Tropical Crops, Ministry of Agriculture, Rubber Research Institute, Chinese Academy of Tropical Agricultural Sciences, Danzhou 571737, China; an-f@catas.cn (F.A.); bqchen@catas.cn (B.C.)

[4] Advanced Analysis and Testing Centre, Nanjing Forestry University, Nanjing 210037, China; uav@njfu.edu.cn

\* Correspondence: xuelianfeng@njfu.edu.cn; Tel.: +86-25-8542-7877

† These authors contributed equally to this work.

**Abstract:** Rubber trees in southern China are often impacted by natural disturbances, and accurate rubber tree crown segmentation and property retrieval are of great significance for forest cultivation treatments and silvicultural risk management. Here, three plots of different rubber tree clones, PR107, CATAS 7-20-59 and CATAS 8-7-9, that were recently impacted by hurricanes and chilling injury were taken as the study targets. Through data collection using ground-based mobile light detection and ranging (LiDAR) technology, a weighted Rayleigh entropy method based on the scanned branch data obtained from the region growing algorithm was proposed to calculate the trunk inclination angle and crown centre of each tree. A watershed algorithm based on the extracted crown centres was then adopted for tree crown segmentation, and a variety of tree properties were successfully extracted to evaluate the susceptibility of different rubber tree clones facing natural disturbances. The results show that the angles between the first-order branches and trunk ranged from 35.1–67.7° for rubber tree clone PR107, which is larger than the angles for clone CATAS 7-20-59, which ranged from 20.2–43.2°. Clone PR107 had the maximum number of scanned leaf points, lowest tree height and a crown volume that was larger than that of CATAS 7-20-59, which generates more frontal leaf area to oppose wind flow and reduces the gaps among tree crowns, inducing strong wind loading on the tree body. These factors result in more severe hurricane damage, resulting in trunk inclination angles that are larger for PR107 than CATAS 7-20-59. In addition, the rubber tree clone CATAS 8-7-9 had the minimal number of scanned leaf points and the smallest tree crown volume, reflecting its vulnerability to both hurricanes and chilling injury. The results are verified by field measurements. The work quantitatively assesses the susceptibility of different rubber tree clones under the impacts of natural disturbances using ground-based mobile LiDAR.

**Keywords:** tree crown segmentation; ground-based mobile LiDAR; rubber tree properties retrieval; natural disturbance

## 1. Introduction

Rubber trees (*Hevea brasiliensis*) are a widely planted hardwood genus in tropical areas and are important suppliers of natural rubber and wood. Hainan Island, which is the largest rubber cultivation base in China, grew approximately $5.4 \times 10^5$ ha of rubber trees in 2015, occupying 16.4% of the total land area of the island and forming the largest artificial ecosystem on the island. However, the hurricanes that occur in the northwestern Pacific Ocean are a major source of disturbance to the Hainan Island rubber tree plantations. The dynamic mechanical loading on rubber trees induced by hurricanes causes windthrow, partial or total defoliation, branch and trunk breakage or inclined trunks with tilting tree bodies. Hence, the accurate acquisition of these rubber tree parameters is an indispensable component of forest cultivation practices and quantitative assessments of the impact of hurricane disturbances on the different rubber tree clones.

Forest parameters, such as tree location, tree height, crown width, diameter at breast height (DBH) and branch angle, are essential for forest management and evaluating different responses to wind loading. Rubber tree structure parameters have been traditionally acquired through field measurements; however, this process is very time consuming, labour intensive and destructive [1] and is useful at only the plot level [2]. Manual measurements are difficult to perform in the harsh conditions of tropical forests, where insect bites and high temperatures are common. Fortunately, the efficiency of manual measurements can be increased by the light detection and ranging (LiDAR) method, which has become one of the most efficient remote sensing technologies for acquiring forest point cloud data with high precision [3,4]. LiDAR is an active remote sensing technology that emits laser pulses to the surface of vegetative elements and analyses the return signal [5]. The point cloud data obtained from LiDAR are valuable and convenient for estimating a variety of tree attributes, e.g., leaf area [6], phenotypic characteristics of leaf elements [7], tree structural attributes [8] and volume of timber [9]. In recent years, the efficiency of measurements has been greatly improved by mobile LiDAR, such as LiDAR loaded on ground vehicles [10], humans (hand-held [11] and backpack [12] modes) and manned aircraft. LiDAR sensors loaded on ground vehicles and humans provide a bottom-up perspective of the high-density representation of tree trunk branches and vegetative elements at the low and middle parts of the forest canopy. Airborne LiDAR [13] provides a top-down measurement setup for the quantitative acquisition of features in the upper tree canopy, such as tree top locations and tree crown attributes; however, the tree properties at low heights are almost completely missed because the laser beam is intercepted by the foliage in the upper forest canopy. Hence, different scanning patterns have unique advantages in terms of providing useful tree characteristics from different scanning angles, which is suitable for different scale areas and forests composed of various tree species.

Accurate crown segmentation from mobile LiDAR point clouds is an essential prerequisite for forest measurements and applications. Tree crown segmentation algorithms have rapidly developed in recent decades, but most of these algorithms are based on airborne LiDAR data. Conventionally, trees are detected from point cloud features and the canopy height model (CHM). A series of point features, such as point density [14], geometrical properties [15] and the spatial distribution of scanned points [16], have been employed to recognize individual tree crown models from a vast amount of scanned data. Continuing efforts using pattern recognition algorithms, such as the mean-shift algorithm [17], K-means clustering [18], region growing [19] and watershed algorithm [20], have also been adopted to accomplish tree crown segmentation based on the detected tree top locations. In addition, other concepts derived from computer science, such as voxelization [21], graph cut algorithms [22], adaptive size window filtering [23], the multilevel morphological active contour method [24], wavelet transform regarding time-frequency decomposition [25] and the topological relationship analysis method [26], have also been extended to delineate tree crowns from airborne LiDAR data. Nevertheless, the segmentation results for these algorithms are always impacted by the accuracy of the tree top detection, the degree of convexity of the top tree crown and the intersected tree crown resulting from tree crown competition.

Accurate individual tree crown segmentation using ground-based mobile LiDAR is still challenging, especially for ecological forests in which tree crowns can be extremely irregular and are often heavily intersected. Although a few pioneering studies on the detection of tree trunk locations for tree segmentation from ground-based mobile LiDAR data have been reported, two separate issues have arisen concerning (1) the deformation of the wood component of the studied trees induced by exposure to perennial hurricane disasters and (2) the deficiency of local scanned data due to self-occluded vegetative elements in lush forests. Aerodynamic drag is known to counter wind-induced tree displacement and causes leaning tree bodies and unevenly distributed defoliation phenomena among rubber forests. The deficiency of local scanned data makes the distribution of scanned points discontinuous, i.e., difficulties in adopting a region growing method based on the scanned trunk points to determine the architecture of the whole tree branch. These disturbances result in difficulties in detecting the tree crown centre and unclear representation of the tree crown shape, which also complicates the delineation of rubber tree crowns from ground-based mobile LiDAR data.

In face of the above issues, this paper proposed a new approach for delineating individual rubber tree crowns and effectively retrieving the tree parameters from mobile ground-based scanned data. This method was used to quantitatively assess the severity of the impacts of wind disturbances on different rubber tree clones, including PR107, CATAS 7-20-59 and CATAS 8-7-9. Due to the biological characteristics of rubber trees, including a strong capacity for water absorption that results in a vegetation-free sub-canopy, the scanned points of the lower part of the rubber tree trunk are easily acquired without occlusion interference and taken as the seed points. Combined with other branch points obtained from the region growing method based on the seed points, a weighted Rayleigh entropy method is proposed to estimate the inclination angle of each rubber tree and locate the centre of each tree crown. Based on the located tree crown centre, a watershed algorithm was adopted to successfully segment the tree crown. Then, the segmented scanned points of each tree were adopted to retrieve a series of tree attributes. Several valuable conclusions are presented based on the comparison of these retrieved tree attributes among the three rubber tree clones under natural disturbance regimes with various severities.

## 2. Materials and Methods

### 2.1. Study Area

The study area was located within a rubber tree plantation in the city of Dan Zhou (northwestern Hainan Island, 109°430–109°510E; 19°280–19°380N, shown in Figure 1). As China's largest rubber production base, the cultivation of rubber trees is continuously increasing in Hainan Island. The topography of the plantation is typically characterized by a hilly plateau with an elevation of 188 m above sea level at the centre. The plateau is surrounded by flat lands with elevations of 20–160 m. Because of sunny and tropical weather with monsoons, the climate is favourable for agricultural development. The annual precipitation is 1600 mm. The rainy season (May–October) accounts for >89% of the total yearly rainfall, and hurricanes of various scales occur during this same period. The mean annual temperature, highest monthly average temperature (June–July), and lowest monthly average temperature (January) are 22.9, 28.0, and 16.9 °C, respectively. The plantation has reclaimed over 5000 ha of cultivated land and tropical rainforest since it was established in 1957. Of these lands, nearly 3000 ha was planted with rubber trees, including a variety of rubber tree clones, including PR107, CATAS7-33-97, CATAS7-20-59, Wenchang217, Haiken2 and CATAS 8-7-9. Over the past sixty years, hurricanes have hit Hainan 101 times. In 2016, tropical storm Dianmu (18th August), severe tropical storm Mirinae (26th July) and super hurricane Sarika (18th October) caused extensive wind damage in the rubber plantations and led to relatively low temperatures, i.e., nearly three months (June–August) with average temperatures of 10 °C. Three tree clones, including rubber tree PR107, rubber tree CATAS 7-20-59 and CATAS 8-7-9, in the rubber tree plantations were chosen as the typical trees for our experiments. Prior knowledge on the cultivation of these rubber trees revealed that some

rubber tree clones (PR107) are more susceptible and vulnerable to wind damage than other rubber tree clones (CATAS7-20-59). The rubber tree clone CATAS 8-7-9 is vulnerable to both hurricane damage and chilling injury; the universal phenomenon of defoliation in CATAS 9-7-9 continues mainly due to chilling injury in the last three months of the year.

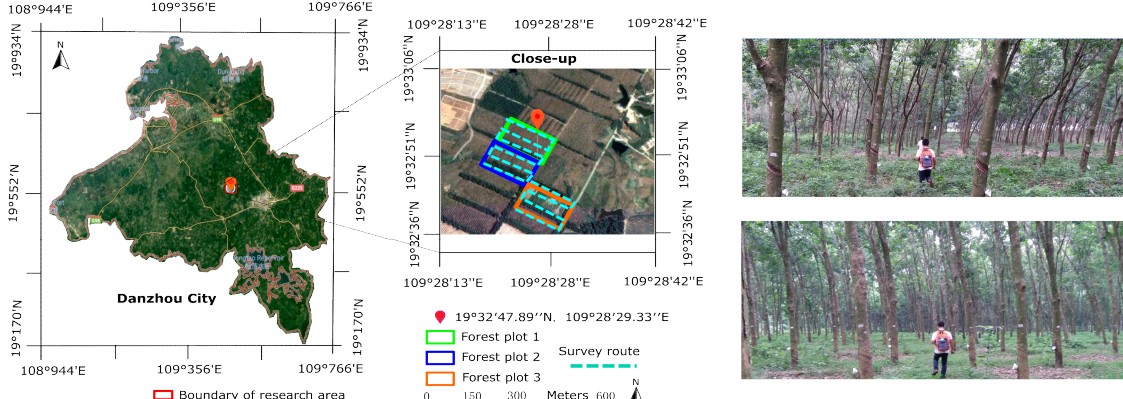

**Figure 1.** Location of the study area and the 3 forest plots of rubber trees within the CATAS experimental farm, Danzhou, Hainan Island, China. The backgrounds of the left and middle picture are the remote sensing image acquired from Google Earth, where the different coloured rectangles mark the edges of the different rubber tree plots and the baby blue dashed lines represent the survey routes using man-portable LiDAR. The photos on the right side show our scanning process in the rubber tree plots using man-portable mobile LiDAR.

### 2.2. Laser Data Acquisition

The LiDAR data were obtained on 10th November 2016, using a Velodyne HDL-32E high-definition LiDAR sensor operable in backpack mode. The sensor has 32 laser/detector pairs that measure the rubber tree plots with the following set parameters: +10.67° to −30.67° vertical field of view (FOV) with an angular resolution of 1.33°, 360° horizontal FOV with an angular resolution of 0.16°, 10 HZ frame rate and 70 m measurement range. The Velodyne HDL-32E scanning system was carried by an experimenter, and the scanner was set to "continuous shooting mode" to collect data at 10 revolutions per second. The experimenter loaded the Velodyne laser scanning system and travelled within the three rubber tree plots according to the predefined survey route. The survey route was programmed to a predefined rectangle parallel plan designed to cover the three study sites. Meanwhile, the experimenter traced survey lines (baby blue dashed lines in Figure 1) at a speed of 0.5 m/s due to the complex terrain of the rubber tree plantation and the heavy scanning instrument. The Velodyne LiDAR system integrates laser scanning with simultaneous localization and mapping (SLAM) technologies to rapidly finish the registration of each scan and generate high-density point clouds for each target rubber tree plot. The mean resolution of the acquired LiDAR data for the three rubber tree plots is approximately 0.02 m.

### 2.3. Field Data

A variety of tree properties, such as tree height, leaf area index (LAI), diameter at breast height (DBH), crown width and the included angle between the trunk and first-order branches, are different for the three rubber tree clones (PR107, CATAS7-20-59 and CATAS8-7-9). Three subsets from the study site of three rubber tree plots were created for detailed testing. Each subset consisted of an approximately 0.6 × 0.6 km area that was representative of the study site. Field measurements within the three subsets were conducted on 11th February 2016. The tree top height was measured using a Vertex IV hypsometer (Haglöf, Långsele, Sweden). The crown widths were obtained as the average of two values measured along two perpendicular directions from the location of the tree top. The included angles between the trunk and first-order branches of all trees located within the subsets were manually measured using a protractor. In situ stem diameter measurements of all rubber trees within the subsets

were collected using a traditional Diameter Tape placed at a height of 1.37 m above the soil surface on the uphill side of the stem. The LAI values of the three rubber tree clone plots were measured using an LAI-2000 plant canopy analyser combined with an optimized sampling strategy for the forest by standardizing the distance and orientation of the LAI-2000 measurements [27]. All these endeavours led to accurate in situ field-based measurement data for the validation of our calculated results.

## 2.4. Pre-Processing of the Scanned Data

The undulating ground of the study area causes inconsistencies in the ground elevation, which results in negative effects on the detection of rubber tree trunks. To eliminate the height inconsistencies of the scanned rubber trees, a program was written by us in Matlab$^@$ to create the digital terrain model (DTM) from the scanned points. The program employed a moving detection window for all scanned data to retrieve the local minimal z value to generate the DTM. Then, the z values of the scanned points of the rubber trees in each window subtract the corresponding DTM value. These raw scanned points are then continuously filtered to eliminate surface height inconsistencies using a customized moving window filter. After the filtering process, the preliminary scanned points of two rubber tree plots on the same height surface were then obtained.

Wood-leaf separation, which aims to classify LiDAR points into wood and leaf components, is an essential prerequisite for determining the branch architecture of a tree and deriving specific leaf characteristics. A definite description of the wood components likely assists with the determination of the canopy centre and degree of plant damage under the impact of a hurricane. Based on the wood-leaf separation algorithm [28], a variety of features for each scanned point were calculated, including the normal vector, the structure tensor and the distribution of the point normal vector. Then, the Gaussian classifier was employed to process the scanned points $P^k$ of the $k$th tree to obtain the separation results of the leaf point set $P_L^k$ and wood point set of each tree. Figure 2 shows the magnified wood-leaf separation results of several typical trees belonging to the three rubber tree clones.

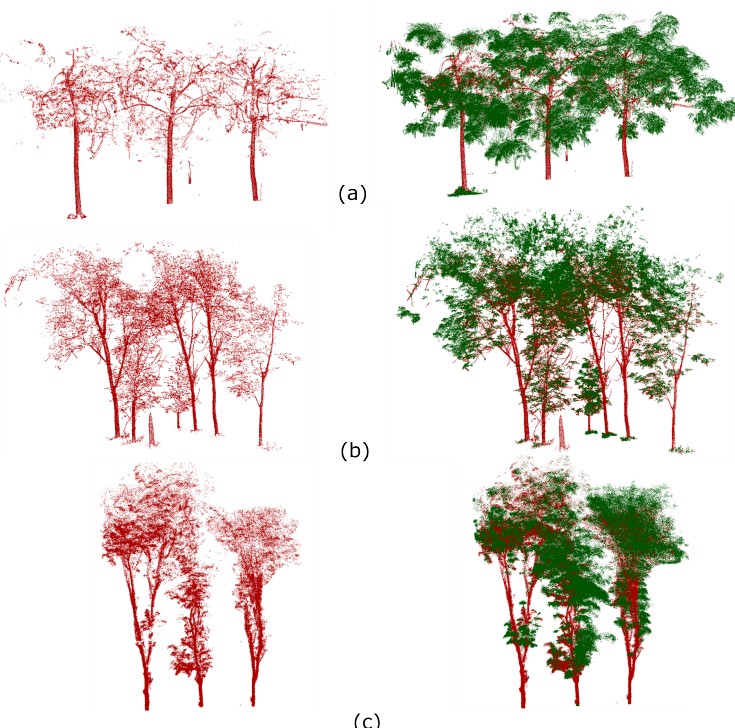

(a)

(b)

(c)

**Figure 2.** Diagram of the wood-leaf classification for the scanned data for several typical trees belonging to the (**a**) PR107, (**b**) CATAS 7-20-59 and (**c**) CATAS 8-7-9 clones, where the green areas represent the classified leaf points, and the crimson areas represent the classified wood points.

## 2.5. *Positioning the Centre of the Tree Crown*

Due to the long-term severe wind disturbances on these rubber trees, mechanical loading on the plants caused by hurricanes results in the inclination of most parts of the rubber tree trunks. Hence, the location of the tree crown centre for each tree cannot be determined directly from the position of the corresponding trunk. Rayleigh entropy was used to calculate the inclination angle of each trunk to derive the centre of each tree crown, which is taken as the primary information for tree crown delineation. The calculation of the inclination angle of each tree trunk is converted to the determination of the directional vector of each trunk. Due to the strong competition by rubber trees, which affects the soil fertility and moisture, the understory within the three rubber tree plots was almost free from sub-canopy vegetation. Hence, the scanned points $p_{i,H/20}^k$, $i \in (1, 2, \dots m)$ on the branch between the ground and one-twentieth of the rubber tree height $H$ can be easily extracted from the wood point set $P_W$. For these points $p_{i,H/20}^k$, $i \in (1, 2, \dots m)$, $m$ represents the total number of the points and $p_{i,H/20}^k$ belongs to the $k$th rubber tree, and these $m$ points are taken as the seed points for the following region growing method [29]. Consequently, these seed points in combination with the region growing method were adopted to iteratively determine whether other branch points were related to the seed points, i.e., searching for connected points based on continuities in distance properties and guaranteeing a point-to-point distance smaller than the threshold $\xi$. An iterative process continues in the same manner until there are no changes in the number of searched points related to seed points between two successive iterative stages. Combined with other branch points $p_j^k$, $j \in (1, 2, \dots n)$ and $p_j^k \in P_W^k$, which were obtained through the region growing method, a fitting strategy based on weighted Rayleigh entropy was adopted to form the $m$ spatial lines for each seed point $p_{i,H/20}^k$ to assess the direction vector $\overrightarrow{v}_i^k (v_{i,x}^k, v_{i,y}^k, v_{i,z}^k)$ of each trunk. For these points $p_j^k$, $j \in (1, 2, \dots n)$, $n$ represents the total number of scanned branch points derived using the region growing method based on the seed points. The concept of our algorithm is illustrated in Figure 3.

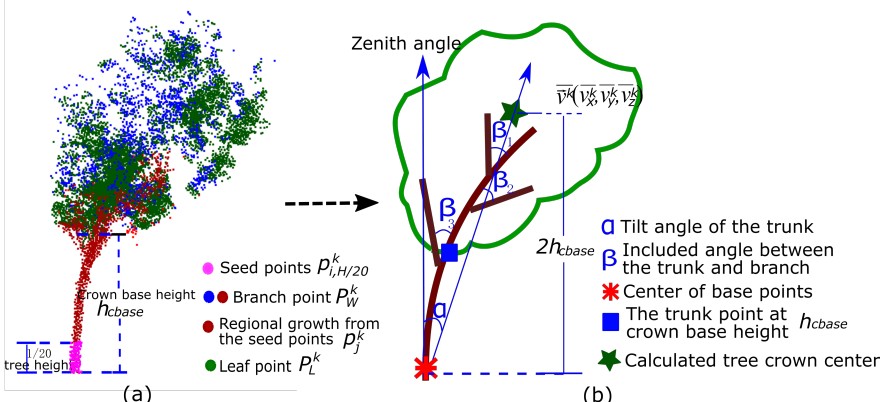

**Figure 3.** Schematic representation of the concept of deriving the location of each tree crown centre. (**a**) The trunk points between the ground and 1/20 of the tree height were chosen as the initial seed points and are represented in pink; the other branch points related to the seed points were found using the region growing method and are represented in brown. The rest of the branch points in blue cannot successfully expand due to occlusion, which causes data discontinuity, which hampers the use of the region growing method from the seed points. (**b**) Based on the points in brown, a weighted Rayleigh entropy-based method was adopted to extract the canopy centre in combination with the centre of the seed points and the trunk point at the crown base height to evaluate the inclination angle of the tree trunk caused by hurricane impacts.

The specific weighted Rayleigh entropy equation used to retrieve the direction vector of each trunk is as follows:

$$\text{argmin}\left(\overrightarrow{v}_i^{\,k}(v_{i,x}^k, v_{i,y}^k, v_{i,z}^k)\right) = \sum_{j=1}^{n} \frac{\varepsilon_j^2 \left|p_{i,H/20}^k - p_j^k\right|^2 \left|\overrightarrow{v}_i^{\,k}\right|^2 - \left(\varepsilon_j\left(p_{i,H/20}^k - p_j^k\right)\cdot \overrightarrow{v}_i^{\,k}\right)^2}{\left|\overrightarrow{v}_i^{\,k}\right|^2} \tag{1}$$

The higher the height, the more first-order and second-order branches exist in the tree crown. The overall orientations of the scanned branch points in the upper tree crown are always consistent with the trunk direction, but the details of each branch direction vary widely, i.e., nearly horizontal, vertical or diagonal. Hence, a weight $\varepsilon_j$ is assigned to each $p_j^k$ according to its $z$ value. The larger the $z$ value of $p_j^k$, the smaller the magnitude of the weight it is assigned to. Hence, the weight $\varepsilon_j = \left(H - z_{p_{i,H/20}^k} - z_{p_j^k}\right)/\left(H - z_{p_j^k}\right)$, where $H$ represents the tree height. Assume that $w_{ij} = \varepsilon_j\left(p_{i,H/20}^k - p_j^k\right)$ and $c_{ij}^2 = \left|w_{ij}\right|^2$; then, Equation (1) can be converted to the following:

$$\text{argmin}\left(\overrightarrow{v}_i^{\,k}(v_{i,x}^k, v_{i,y}^k, v_{i,z}^k)\right) = \frac{\sum_{j=1}^{n} c_{ij}^2 (\overrightarrow{v}_i^{\,k})^T \overrightarrow{v}_i^{\,k} - (v_{i,x}^k w_{ij,x} + v_{i,y}^k w_{ij,y} + v_{i,z}^k w_{ij,z})^2}{(\overrightarrow{v}_i^{\,k})^T \overrightarrow{v}_i^{\,k}} = \frac{\alpha(\overrightarrow{v}_i^{\,k}) - \beta(\overrightarrow{v}_i^{\,k})}{(\overrightarrow{v}_i^{\,k})^T \overrightarrow{v}_i^{\,k}} \tag{2}$$

where $\alpha\left(\overrightarrow{v}_i^{\,k}\right) = (\overrightarrow{v}_i^{\,k})^T \overrightarrow{v}_i^{\,k} \sum_{j=1}^{S} c_{ij}^2$ and

$$\beta\left(\overrightarrow{v}_i^{\,k}\right) = (v_{i,x}^k)^2 \sum_{j=1}^{n} w_{ij,x}^{\,2} + (v_{i,y}^k)^2 \sum_{j=1}^{n} w_{ij,y}^{\,2} + (v_{i,z}^k)^2 \sum_{j=1}^{n} w_{ij,z}^{\,2} + 2v_{i,x}^k v_{i,y}^k \sum_{j=1}^{n} w_{ij,x}w_{ij,y} +$$

$$2v_{i,x}^k v_{i,z}^k \sum_{j=1}^{n} w_{ij,x}w_{ij,z} + 2v_{i,y}^k v_{i,z}^k \sum_{j=1}^{n} w_{ij,y}w_{ij,z}$$

$$\text{Assume that } U = \begin{pmatrix} w_{i1}^T \\ w_{i2}^T \\ \vdots \\ w_{in}^T \end{pmatrix} = \begin{pmatrix} \varepsilon_1\left(p_{i,H/20}^k - p_1^k\right) \\ \varepsilon_2\left(p_{i,H/20}^k - p_2^k\right) \\ \vdots \\ \varepsilon_n\left(p_{i,H/20}^k - p_n^k\right) \end{pmatrix} \tag{3}$$

We defined covariance matrix $B = U^T U$, where the size of $U$ is $n \times 3$ and the size of covariance matrix $B$ is $3 \times 3$. Then, $\alpha$ and $\beta$ can be expressed as $\alpha\left(\overrightarrow{v}_i^{\,k}\right) = (\overrightarrow{v}_i^{\,k})^T (el)\overrightarrow{v}_i^{\,k}$ and $\beta\left(\overrightarrow{v}_i^{\,k}\right) = (\overrightarrow{v}_i^{\,k})^T B \overrightarrow{v}_i^{\,k}$, where $e = \sum_{j=1}^{n} c_{ij}^2$ and $l$ is a $3 \times 3$ unit matrix. Then, Equation (2) can be written as follows:

$$\text{argmin}\left(\overrightarrow{v}_i^{\,k}(v_{i,x}^k, v_{i,y}^k, v_{i,z}^k)\right) = \frac{(\overrightarrow{v}_i^{\,k})^T (el)\overrightarrow{v}_i^{\,k} - (\overrightarrow{v}_i^{\,k})^T B \overrightarrow{v}_i^{\,k}}{(\overrightarrow{v}_i^{\,k})^T \overrightarrow{v}_i^{\,k}} = \frac{(\overrightarrow{v}_i^{\,k})^T (el - B)\overrightarrow{v}_i^{\,k}}{(\overrightarrow{v}_i^{\,k})^T \overrightarrow{v}_i^{\,k}} = \frac{(\overrightarrow{v}_i^{\,k})^T M \overrightarrow{v}_i^{\,k}}{(\overrightarrow{v}_i^{\,k})^T \overrightarrow{v}_i^{\,k}} \tag{4}$$

where $M = el - B$. According to the mathematical property of the weighted Rayleigh entropy, the value of Equation (4) reaches the minimal value when $\overrightarrow{v}_i^{\,k}$ equals the eigenvector corresponding to the minimal eigenvalue of matrix $M$. Using the feature decomposition theorem of matrix theory, we obtained the following equation:

$$M\xi = (el - B)\xi = el\xi - B\xi = e\xi - \lambda\xi = (e - \lambda)\xi \tag{5}$$

where $\lambda$ and $\xi$ represent the eigenvalue and eigenvector of matrix $B$, respectively. As shown in Equation (5), a close relationship exists between the eigenvalues of matrix $M$ and $B$. The search for the eigenvector corresponding to the minimal eigenvalue of $M$ can be converted to finding the eigenvector

corresponding to the maximal eigenvalue of $B$. Hence, the eigenvector corresponding to the maximal eigenvalue of $B$ was taken as the directional vector of each trunk. Then, based on the combination of each seed point $p^k_{i,H/20}$ and the corresponding retrieved directional vector $\overrightarrow{v}^k_i$, each fitted line was formed to guarantee the minimal orthogonal distance between the trunk points and the spatial line. The base fitting straight line of the $i$th base point can be expressed as follows:

$$\frac{x - p^k_{i,H/20,x}}{v^k_{i,x}} = \frac{y - p^k_{i,H/20,y}}{v^k_{i,y}} = \frac{z - p^k_{i,H/20,z}}{v^k_{i,z}} \tag{6}$$

Based on the above methods, we iteratively chose every seed point $p^k_i$, $i \in (1, 2, \ldots m)$ belonging to each tree and calculated the corresponding directional vector $\overrightarrow{v}^k_i (v^k_{i,x}, v^k_{i,y}, v^k_{i,z})$. Then, the average value of all seed points of each tree and the average directional vector can be calculated and expressed as $\overline{p^k}(\overline{x^k_{avg}}, \overline{y^k_{avg}}, \overline{z^k_{avg}})$ and $\overline{\overrightarrow{v}}^k (\overline{v^k_x}, \overline{v^k_y}, \overline{v^k_z})$, respectively.

Hence, the optimal fitting point-slope equation $\overline{L^k}$ for each rubber tree trunk can be expressed as follows:

$$\frac{x - \overline{x^k_{avg}}}{\overline{v^k_x}} = \frac{y - \overline{y^k_{avg}}}{\overline{v^k_y}} = \frac{z - \overline{z^k_{avg}}}{\overline{v^k_z}} = t \tag{7}$$

Meanwhile, according to empirical knowledge, the intersection point of the fitting line at twice the tree crown base height $z = 2h_{cbase}$ is close to the canopy centre. Consequently, the coordinates of the canopy centre of the $k$th rubber tree were obtained as follows:

$$\left( \frac{(2h_{cbase} - \overline{z^k_{avg}})\overline{v^k_x}}{\overline{v^k_z}} + \overline{x^k_{avg}}, \; \frac{(2h_{cbase} - \overline{z^k_{avg}})\overline{v^k_y}}{\overline{v^k_z}} + \overline{y^k_{avg}}, \; 2h_{cbase} \right) \tag{8}$$

The average branch inclination angle of each tree trunk can be calculated by calculating the included angle between $\overline{\overrightarrow{v}}^k$ and the zenith direction $\overrightarrow{v}_2(0, 0, 1)$. The equation is as follows:

$$\alpha^k = \arccos \frac{\overline{\overrightarrow{v}}^k \cdot \overrightarrow{v}_2}{\left|\overline{\overrightarrow{v}}^k\right|\left|\overrightarrow{v}_2\right|} = \arccos \frac{\overline{v^k_z}}{\sqrt{\overline{v^k_x}^2 + \overline{v^k_y}^2 + \overline{v^k_z}^2}} \tag{9}$$

## 2.6. Tree Crown Delineation Based on the Watershed Algorithm

Based on the calculated centre of each tree crown, all of the scanned data from the three rubber tree plots were vertically projected onto a plane. The lack of obvious convex shape for the upper part of the rubber tree crowns precludes the occurrence of adjacent catchment basins for watershed processing and prevents the collection of continuous image gradient information to conveniently form contours that delineate each tree crown. Hence, a watershed segmentation method based on the Euclidean distance metric [30] based on the extracted each tree crown centre was adopted here to obtain the boundaries of each tree crown, and the obtained boundaries are shown in Figure 4. Then, the projected status was reverse transformed into the original scanned points, and the tree crown boundaries obtained from the watershed algorithm assisted with the segmentation of each tree crown. Consequently, a variety of individual tree properties were retrieved based on the separated scanned points from individual trees. The volume of each tree crown was assessed using the convex hull algorithm [31] based on the separated leaf points of each tree. A cylinder model was adopted to fit the trunk, and the diameter of the cylinder represents the DBH of each tree. The tree model reconstructing method [32] was used to define the stretching direction of the branches capable of retrieving the angle $\beta$ between the first-order

branches and trunk. The calculated tree parameters were compared with field measurements to verify the effectiveness of our method.

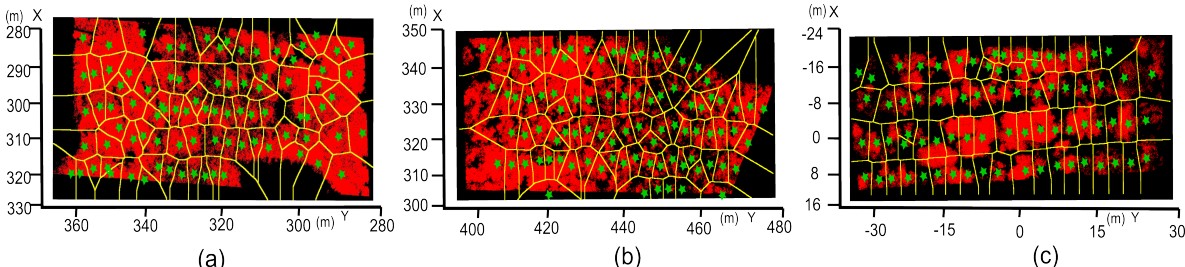

(a)                  (b)                 (c)

**Figure 4.** Extracted boundaries of tree crowns using the watershed algorithm based on the tree crown centre and Euclidean distance. (**a**) Rubber tree plot 1 (PR107), (**b**) Rubber tree plot 2 (CATAS 7-20-59) and (**c**) Rubber tree plot 3 (CATAS 8-7-9). The segmentation strategy is preferred here for rubber tree plantations with similar silvicultural treatments and forest structural parameters.

## 3. Results

The wood-leaf separation results for the scanned points of subsets from three rubber tree plots are shown in Figure 5. Tree branches are always located in the intermediate of the tree crown, and self-occluded vegetation elements in rubber tree plots generally obstruct laser scanning views, which results in the incompleteness of some scanned branch and leaf data. According to the criteria of the region growing method, i.e., searching the branch points that were related to seed points and satisfying a point-to-point distance smaller than the threshold $\xi$, complete branch information could not be obtained for some of the trees due to occlusion, which caused the deficiency of some of the scanned branch data. However, most of the branch points in the lower and middle parts of the rubber trees in the three plots were well obtained using the region growing method (Figure 6). Consequently, combined with weighted Rayleigh entropy and branch points obtained from seed points, the directional vector of the fitted line depicting the degree of inclination of the trunk of each tree was assessed. Based on silvicultural knowledge, the intersection points of every fitted line at approximately twice the tree crown base height $z = 2h_{cbase}$ were taken as the canopy centre of each tree, which is marked by a green pentagram in Figure 7. Then, based on the position of the retrieved centre of each tree crown, the vertical projection of scanned points for the three rubber tree plots was determined in combination with the watershed algorithm [20] to achieve the tree crown segmentation results. The tree crown segmentation results are shown in Figure 8.

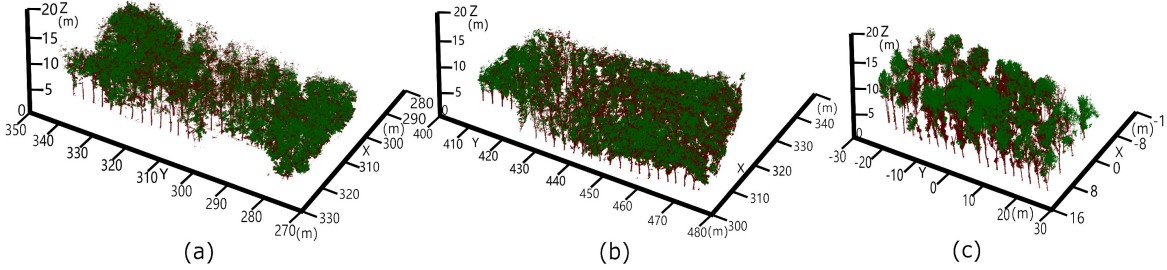

(a)                  (b)                 (c)

**Figure 5.** Program diagrams showing our wood-leaf separation results based on the scanned data, where the scanned branch points are indicated in brown, and the scanned leaf points are indicated in green. (**a**), (**b**) and (**c**) represent the segment results for the subset from rubber tree plots 1 (PR107), 2 (CATAS 7-20-59) and 3 (CATAS 8-7-9), respectively.

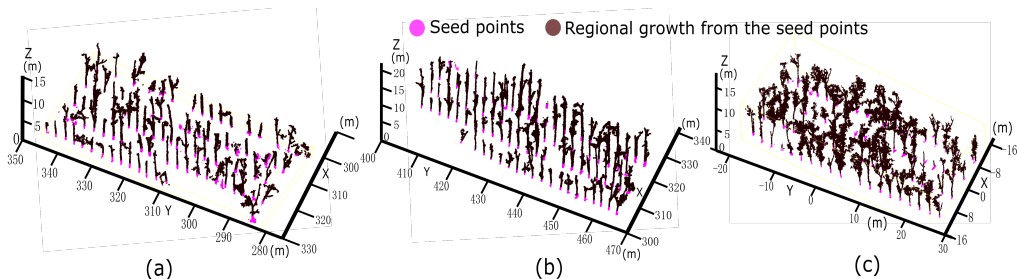

**Figure 6.** Program diagrams showing the results of the regional growth from the seed points, where the branch points of each tree related to the corresponding seed points were extracted and represented in brown. (**a**), (**b**) and (**c**) show the region growing results for the subsets from rubber tree plot 1 (PR107), rubber tree plot 2 (CATAS 7-20-59) and rubber tree plot 3 (CATAS 8-7-9), respectively.

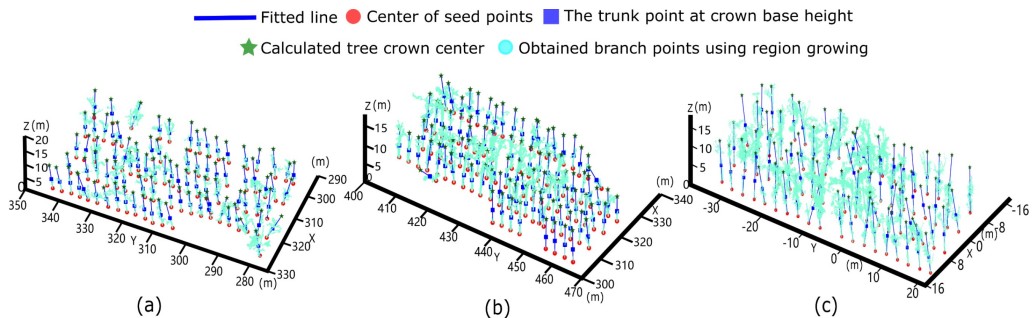

**Figure 7.** Based on the calculated branch points of each tree using the region growing method, Rayleigh entropy was adopted to derive the directional vector of the fitted line for the trunk structure of each tree. Combined with tree height and empirical knowledge, the intersection points at twice the tree crown base height were taken as the crown centre of each tree, which is marked with a green pentagram. (**a**), (**b**) and (**c**) show the corresponding results for the subset from rubber tree plot 1 (PR107), rubber tree plot 2 (CATAS 7-20-59) and rubber tree plot 3 (CATAS 8-7-9), respectively.

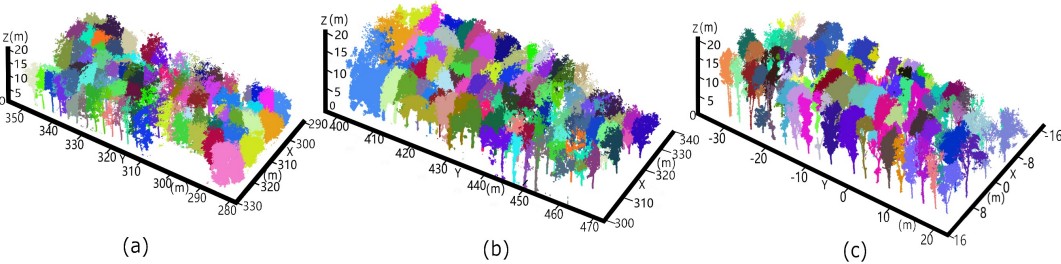

**Figure 8.** Tree crown segmentation results using a watershed algorithm based on the detected tree crown centres, where different colours indicate different trees. (**a**), (**b**) and (**c**) show the corresponding results for the subset from rubber tree plot 1 (PR107), rubber tree plot 2 (CATAS 7-20-59) and rubber tree plot 3 (CATAS 8-7-9), respectively.

The quantitative evaluation of the comparison of the different tree property retrieval methods using our method versus the field measurements is listed in Table 1. Under the same scanning resolution and approximately the same planting spacing for each rubber tree plot, the retrieved average tree height from the scanned points of rubber tree clone PR107 is lower than that from rubber tree clone CATAS 7-20-59, but more scanned leaf points were obtained for PR107 than CATAS 7-20-59, which means that more leaf elements existed in the tree crown, leading to a higher LAI for rubber tree plot 1 than plot 2. Meanwhile, the angle $\beta$ between the first-order branches and the trunk is largest for PR107 and ranges from 35–68° (Figure 2 shows the topological structures of typical tree skeletons belonging to different clones), thus accounting for a larger tree crown volume with a spread out crown that

provides more space for leaf growth. Nevertheless, this branch architecture increases the instability of tree structures under wind loads. Meanwhile, high LAI values increase the frontal leaf area opposing wind flow, which induces excess loads on the PR107 rubber tree clone by wind gusts and increases the vulnerability of the trees to wind damage. The angle $\beta$ for CATAS 7-20-59 ranges from 20–43° with an average DBH of 33.10 cm, which forms the vase shape of the tree crown and maintains a stable structure. The tree crown volume and leaf area of CATAS 7-20-59 are lower than those of PR107, which causes large gaps among the forest canopy and benefits the passing of air flow through the forest and strengthens the wind resistance. The rubber tree clones CATAS 8-7-9 are susceptible to chilling injury, and severe defoliation phenomena have occurred due to the recent low temperatures. Hence, the minimum average tree crown volume and number of scanned leaf points occur in rubber tree plot 3. Table 1 quantitatively indicates that our calculated results derived from the scanned points adequately fit the field measurements. The radar charts in Figure 9 show the spatial distribution of scanned leaf points of each tree in different rubber tree plots. It is clear that the number of individual tree leaf points of the rubber tree clone PR107 is larger than that of the rubber tree clones CATAS 7-20-59 and 8-7-9, which verifies that the rubber tree clone PR107 has a higher LAI than other rubber tree clones. The box plots in Figure 10 depict the distribution of our calculated tree properties regarding individual rubber trees in rubber tree plots 1, 2 and 3. As shown in Figure 10b, high LAI and large crown volumes result in increased wind loads on trees, which resulted in a larger trunk inclination angle for PR107 (4.3–30.1°) than CATAS 7-20-59 (1.1–18.0°).

**Table 1.** Accuracy of the retrieved parameters for three rubber tree plots using our method in comparison with field measurements.

| | Rubber Tree Plot 1 (PR107) | Rubber Tree Plot 2 (CATAS 7-20-59) | Rubber Tree Plot 3 (CATAS 8-7-9) |
|---|---|---|---|
| Number of scanned points/Number of trees | 1359879/138 | 1325866/148 | 922628/191 |
| Number of scanned points (Leaf/Wood) | 1039143/320736 | 958631/367235 | 660558/262070 |
| Plant spacing (m) | 6.08 Vertical 2.52 Horizontal | 6.13 Vertical 2.52 Horizontal | 6.14 Vertical 2.62 Horizontal |
| Retrieved parameters | (Our method/Field measurement), Correlation degree | (Our method/Field measurement), Correlation degree | (Our method/Field measurement), Correlation degree |
| Average tree height (m) | (13.23/13.17), 96.34% | (14.95/15.10), 98.32% | (12.92/13.11), 97.15% |
| Average breast diameter (cm) | (27.12/26.21), 97.21% | (33.10/33.29), 98.43% | (21.91/21.06), 96.32% |
| Average crown volume (m³) | (205.45/200.98), 92.82% | (182.00/184.48), 91.71% | (99.47/ 103.82), 90.31% |
| Crown length (m) | (3.74/3.95), 93.72% E-W (5.59/5.78), 95.13% N-S | (3.07/3.08), 98.72% E-W (5.48/5.70), 96.12% N-S | (3.92/3.96), 97.81% E-W (4.87/4.69), 96.30% N-S |
| Average trunk inclined angle α (degree) | (13.08°/12.97°), 97.16% | 8.14°/8.49°, 95.92% | 8.87°/9.17°, 94.74% |
| The angle β between the first-order branch and trunk (degree) | (35.06°–67.73°/ 36.62°–66.49°) | (20.29°–43.20°/ 21.34°–41.18°) | (23.14°–54.36°/ 24.28°–55.38°) |

Note: E-W, east-west direction; N-S north-south direction.

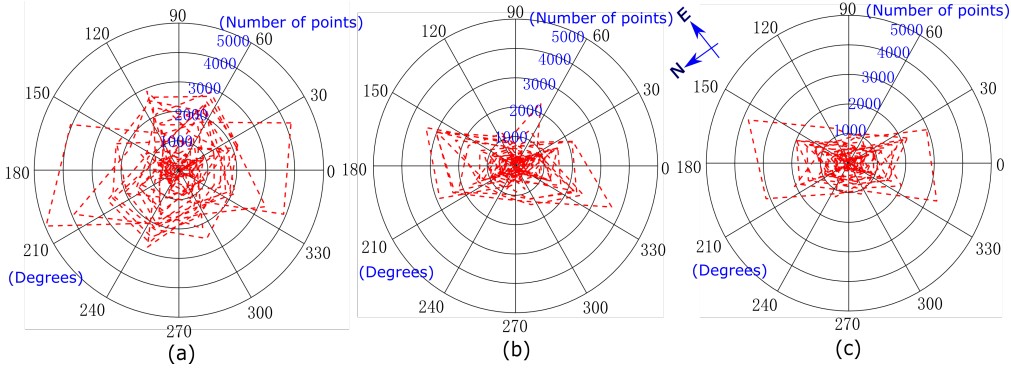

**Figure 9.** Radar charts showing the spatial distribution of leaf points for individual trees in rubber tree plot 1 (**a**), plot 2 (**b**) and plot 3 (**c**). The spokes of the radar maps indicate that the number of scanned leaf points in different directions for rubber tree plot 1 is larger than that for rubber tree plots 2 and 3.

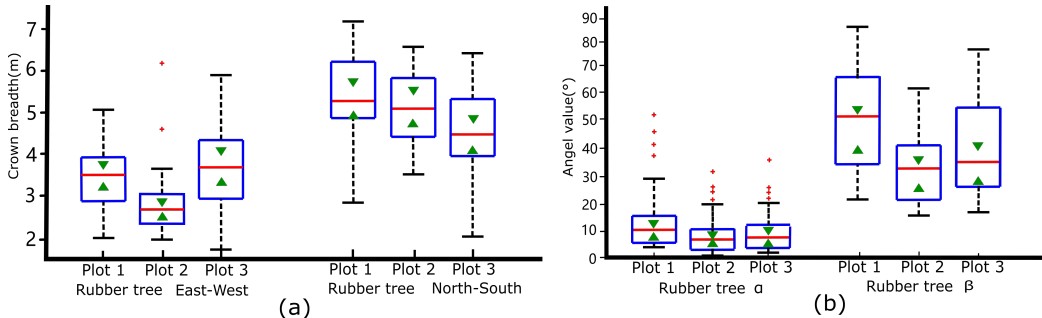

**Figure 10.** Box plots depicting the parameter distribution retrieved using our algorithm from sixty randomly selected trees in each rubber tree plot. (**a**) Distribution of the group of calculated results regarding crown breadth in the East-West and North-South directions for the different rubber tree plots. (**b**) Distribution for the group of calculated results regarding the tilt angle of the trunk and the included angle between the trunk and branch for different rubber tree plots.

## 4. Discussion

### 4.1. Specific Uses of Our Approach

There is increasing interest in the accurate estimation of tree properties in planted forests for assessing the efficiency of forest cultivation under different natural environmental influences. Individual tree segmentation is still an essential premise for tree property retrieval from various types of remote sensing data of forests. If the trees that are planted in forests have heavily intersected tree crowns, seriously sloped trunks and unobvious tree crown features, these interference factors complicate the tree crown segmentation based on scanned data. Some researchers have adopted mobile ground-based LiDAR, such as the man-portable backpack scanning mode, to scan the data of tree trunks growing in forests [10] and combined the region growing method based on trunk data to perform individual tree segmentation. Nevertheless, in a lush forest with a high LAI, ubiquitous occlusion effects result in considerable deficiencies of local scanned data, which seriously hampers continuous expansion from the trunk points to the upper tree branch structure based on the region growing method. In addition, it is easy to generate fallible expansion directions for heavily intersected tree crowns using the region growing method. In our study, the biological properties of rubber trees, such as the strong capacity for water absorption and high LAI, result in the mortality of underlying vegetation and markedly alleviate the occlusion effect, which allows for the scanned points for the lower wood components of each rubber tree to be fully captured. Based on these scanned points, a weighted Rayleigh entropy method in combination with other branch points in the middle and upper tree crown obtained from the wood-leaf separation algorithm was used to derive the direction of the overall architecture of the wood components of each tree. The spatial position of each tree crown centre is associated with the overall rubber tree height of different clones, and it is convenient to deduce this information. Consequently, tree crown segmentation was achieved, and various attributes of individual rubber trees belonging to different clones were retrieved to analyse the wind-resistant performances of different clones.

The proposed method is more suitable for processing pure plantations or contiguous areas that contain a number of relatively homogeneous trees or have a common set of growth characteristics, such as tree age [33], tree height and crown width [34,35]. For the study plots with grown trees of different tree species or ages, although the spatial growth direction (normal vector) of each tree trunk is successfully derived using our algorithm from mobile ground-based LiDAR, the variation in tree height results in a non-uniform height distribution of tree crown centre, which invalidates our method of calculation of tree crown centre by searching the intersection point of the fitting line at a fixed tree height (Equation (8)). This work is plausible for the three rubber tree plots in our study because the trees in these plots have similar properties in their composition, crown breadth, age and spatial arrangement. However, for a forest plot that presents inconsistent tree species and crown breadth or

trees that exhibit strong spatial competition and supress the neighbour trees, tree crown segmentation that depends on the Euclidean distance between each tree crown centre will not be reliable. For these cases, other useful information, such as phenotypic features of the tree crown and topological structure characteristics, must be synthetically considered.

### 4.2. Impact of Hurricane Propagation through the Rubber Tree Forests

The rubber tree plots act as a barrier when met with the strong wind flow caused by hurricanes. The weak resistance of the forest canopy cannot stop the wind from moving forward and only decelerates the wind speed at the cost of defoliation and branch breakage. Hence, we calculated the leaf area density (LAD) distribution of the whole forest section according to the scanned point density distribution to analyse the damage resulting from hurricane propagation through the rubber tree plots. As shown in Figure 11, the red dotted boxes indicate the area where a significant decrease in LAD occurred. For rubber tree plots 1 and 2, high LAD values result in small gaps during the initial hurricane stage, which induces the wind to seek a breakthrough path and results in early enhancement to the damage propagation through the edge trees along the woodland path. Therefore, the decreasing LAD phenomena dominate the left side of rubber tree plots 1 and 2, which are close to the woodland path and marked by the left red dotted boxes in Figure 11a,b. With the drastic increase in wind intensity, the wind force is strengthened, and wind rushes through a path near the middle of the rubber tree plots and passes through the plots, which results in defoliation and wood structural fatigue along the path. The red dotted boxes in the middle of Figure 11a,b indicate the wind path with foliage destruction in rubber tree plots 1 and 2. For rubber tree plot 3, a ubiquitous leaf area decrease occurred throughout the forest canopy that was mainly due to the recent spread of chilling injury throughout the plot, which caused increases in the pervasive forest gap sizes, allowing wind gusts to pass smoothly through the plot. Hence, no obvious local LAD reduction existed in this plot.

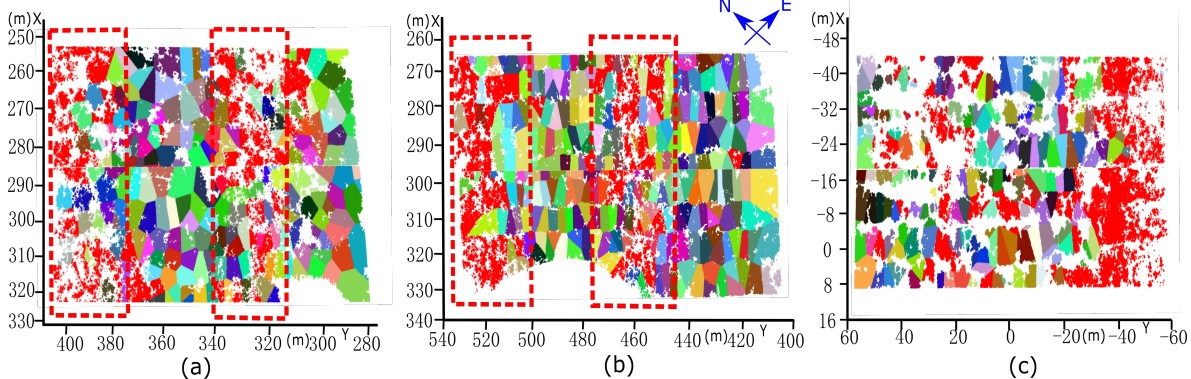

**Figure 11.** After tree crown delineation, the relative LAD was calculated based on the separated scanned leaf points. The dotted red boxes represent the areas with reduced LAD and indicate the path of the hurricane winds through the rubber tree plots, which induced defoliation. (**a**) Rubber tree plot 1 (PR107). (**b**) Rubber tree plot 2 (CATAS 7-20-59). (**c**) Rubber tree plot 3 (CATAS 8-7-9) recently suffered from serious chilling injury, and ubiquitous defoliation spread throughout the plot with large gaps in the forest canopy, which benefited the passing of wind gusts through the plot, leading to no obvious local LAD decreases.

## 5. Conclusions

The quantitative assessment of the susceptibility of different clones of rubber trees to wind damage is urgently needed. For crooked rubber trees caused by long-term hurricane disturbances, a weighted Rayleigh entropy method based on scanned branch points was designed to locate the overall direction of the structure of wood components and accomplish tree crown segmentation. Consequently, a variety of rubber tree properties were retrieved from the segmented scanned points of each tree.

The results show that the average scanned leaf points per tree, average crown volume and the angle between the first-order branch and trunk of PR107 (9854, 205.5 m$^3$ and 13.08°, respectively) are larger than those of CATAS 7-20-59 (8959, 182.0 m$^3$ and 8.14°, respectively), which results in more frontal leaf area and unstable tree structure for wind loading. The prior practical experience of rubber tree silviculture also underlines the conclusion that clone PR107 is more susceptible to wind storms than clone CATAS 7-20-59. Meanwhile, marked decreases in leaf area and crown volume indicate dual disasters stemming from chilling injury and severe hurricanes for clone CATAS 8-7-9. The success of our proposed algorithm provides a solid foundation for the segmentation of rubber tree crowns and the retrieval of rubber tree parameters based on ground-based mobile LiDAR data and provides a quantitative assessment of the forest impacts following a natural disturbance. Further research should consider developing a universal and robust tree crown segmentation algorithm with a variety of tree phenotypic parameters for different types of forests that are impacted by natural disturbances.

**Author Contributions:** Methodology, T.Y. and L.X.; Software, K.J. and H.H.; Validation, A.J. and W.L.; Investigation, F.A. and B.C.; Writing—original draft preparation, T.Y. and K.J.; Writing—review and editing, T.Y. and K.J.; Visualization, K.J. and H.H.

**Funding:** This work was partly supported by the National Natural Science Foundation of China via grant 31770591 and 41701510, the National Key Research and Development Program of China via grant 2017YFD0600905-1, and the Priority Academic Program Development of Jiangsu Higher Education Institutions (PAPD).

**Conflicts of Interest:** The authors declare no conflicts of interest.

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
