# Peer review of "Rubber Tree Crown Segmentation and Property Retrieval Using Ground-Based Mobile LiDAR after Natural Disturbances"

_remotesensing, doi:10.3390/rs11080903_

Round 1

Reviewer 1 Report

Comments

This manuscript deals with crown characteristics segmentation after natural disturbances based on ground-based mobile LiDAR data. The results could provide information concerning crown characteristics rubber trees after natural disturbances. The comments and suggestions are as following.

1. Title and Abstract can reflect whole text that are suitable for audiences.

2. I only have a suggestion for Introduction chapter. Since this paper deals with crown characteristics of individual tree, I provide some international papers as references. Authors could consider to add in this chapter or discussion chapter.

Baldwin Jr.V.C., Peterson K.D., (1997). Predicting the crown shape of loblolly pine tree. Canadian Journal of Forest Research 27: 102–107. DOI: 10.1139/x96-100.

Yen T. M. (2015) Relationships of Chamaecyparis formosensis crown shape and parameters with thinning intensity and age. Ann. For. Res. 58(2): 323-332. DOI: 10.15287/afr.2015.408

3. In Discussion, the limitation of the present technique should be revealed in this chapter as well.

Overall, I believe that this manuscript has contribution in the remote sensing file and I am pleasure to recommend it for publication in the “remote sensing”.

Author Response

The attached word file was uploaded to present the detailed information of the responses to the reviewer's comments.

Reviewer 2 Report

Please find my comments in the attached file.

Author Response

(The authors gave the same response as above.)

Reviewer 3 Report

The paper “Rubber tree crown segmentation and property retrieval using ground-based mobile LiDAR after natural disturbances” by Ting Yun, Kang Jiang, Hu Hou, Feng An, Bangqian Chen, Anna Jiang, Weizheng Li and Lianfeng Xue is devoted to the algorithm created by the authors for modeling of rubber tree forest plots. Modeling is based on a combination of on ground-based mobile LiDAR data and analytical methods. The calculation algorithm and measuring technique are clearly presented in the paper.

The authors proposed a practically useful approach for studying the hurricane and chilling damage influence on LAD distribution in rubber tree crowns. The novelty of study is also clear from the text.

I suppose the paper can be published in Remote Sensing.

My only remark is that all formulas, at list in my version of the paper, are badly printed, that is fine and distorted.

Author Response

(The authors gave the same response as above.)
